# Design and Development of Ti–Ni, Ni–Mn–Ga and Cu–Al–Ni-based Alloys with High and Low Temperature Shape Memory Effects

**DOI:** 10.3390/ma12162616

**Published:** 2019-08-16

**Authors:** Vladimir Pushin, Nataliya Kuranova, Elena Marchenkova, Artemy Pushin

**Affiliations:** 1Laboratory of Non-ferrous Alloys, M.N. Miheev Institute of Metal Physics of Ural Branch of Russian Academy of Sciences, Ekaterinburg 620108, Russia; 2Laboratory of structural and functional steels and alloys, Institute of new materials and technologies, Ural Federal University, Yekaterinburg 620002, Russia

**Keywords:** thermoelastic martensitic transformation, shape memory effect, strength and ductility, TiNi, Ni_2_MnGa, Cu–Al–Ni alloys, structure types, parameters of microstructure, ultra-fine grain size

## Abstract

In recent years, multicomponent alloys with shape memory effects (SMEs), based on the ordered intermetallic compounds B2–TiNi, L2_1_–Ni_2_MnGa, B2– and D0_3_–Cu–Me (Me = Al, Ni, Zn), which represent a special important class of intelligent materials, have been of great interest. However, only a small number of known alloys with SMEs were found to have thermoelastic martensitic transformations (TMTs) at high temperatures. It is also found that most of the materials with TMTs and related SMEs do not have the necessary ductility and this is currently one of the main restrictions of their wide practical application. The aim of the present work is to design and develop multicomponent alloys with TMTs together with ways to improve their strength and ductile properties, using doping and advanced methods of thermal and thermomechanical treatments. The structure, phase composition, and TMTs were investigated by transmission- and scanning electron microscopy, as well as by neutron-, electron- and X-ray diffraction. Temperature measurements of the electrical resistance, magnetic susceptibility, as well as tests of the tensile mechanical properties and special characteristics of SMEs were also used. Temperature–concentration dependences for TMTs in the binary and ternary alloys of a number of quasi-binary systems were determined and discussed. It is shown that the ductility and strength of alloys required for the realization of SMEs can be achieved through optimal alloying, which excludes decomposition in the temperature range of SMEs’ usage, as well as via various treatments that ensure the formation of their fine- (FG) and ultra-fine-grained (UFG) structure.

## 1. Introduction

For more than 50 years, so-called intelligent (or smart) binary and low-doped alloys based on titanium nickelide undergoing reversible thermoelastic diffusionless martensitic transformations (TMTs) attract great attention owing to the best combination of excellent structural and functional properties [1,2,3,4,5,6]. These alloys are distinguished by unusual effects of the thermomechanical shape memory (SMEs), pseudoelasticity (PE) and damping in combination with high-strength and ductile characteristics, excellent long-term strength and durability, corrosion resistance and biocompatibility. These materials are widely used in aerospace, automotive, mining, robotic, biomedical and other fields, including the creation of various micro-electro-mechanical systems (MEMS) such as sensors and actuators [2,3,4,5,6,7,8,9,10].

A unique key feature of the B2-austenite of titanium nickelide alloys in the pre-martensitic state is the total isotropic softening of all the elastic constants Cij, modules E and G, discovered and discussed first in [11,12,13,14,15,16,17,18,19,20,21] (Figure 1). In this case, according to inelastic neutron scattering there is a significant softening of the acoustic transversely polarized phonon modes TA_2_ <ξξ0> *κ* <110> e, especially, in the vicinity of the wave vectors *κ* at ξ = 1/3 and ξ = 1/2, which shows progress when approaching the start temperature M_s_ of TMTs [22,23,24,25]. Anisotropic diffuse scattering of the X-rays, electrons and the striation of tweed image contrast in the transmission electron microscopy (TEM) images are also found (Figure 2) [4,6,15,17]. These anomalous effects are caused by the lattice instability and appearance of localized quasi-elastic phonon domains that form the short order of atomic displacements (SOD). By means of the long-period modulation (LPD) in such quasi-static nanodomains, which is responsible for emerging diffraction satellites of types “1/3” and “1/2”, their structure can be reconstructed into the structure of the martensitic phases R, B19 or B19′, respectively [15,16,17,18,19]. Within the framework of this paradigm of structural-phase transformations, localized heterogeneous nanodomains with SOD and LPD in the pre-transition austenite state are considered not only as nanostructured precursors, but also as real nuclei of thermoelastic martensite crystals of the R, B19 and B19′ types at the stage preceding the TMTs themselves [4,6,17,19]. Following these views, a crystal-geometrical model of structural-phase reconstruction was proposed using the Bain deformation and shuffling shear displacements, for example, of the type {011}<011> for B2→B19 TMTs or combined displacements of the type {011}<100> + {011}<011> for B2→B19′ TMTs (Figure 3). Here, a small magnitude of the volume effect ΔV/V = (V_M_ − V_A_)/V_A_ × 100% is very important.

In fact, this model is consistent with the two-step scenario presented in [26,27,28]. In a first step, B2→B19′ reconstruction occurs through shear on the basal plane (011)[011]. In a second step, B19 transforms to B19′ martensite in a shear process on the non-basal plane (100)[01 1]_B2_. In this second process, the crystal lattice angle changes from 90 to 97 (see Figure 3c).

It should be noted that in other metastable non-ferrous alloys with qualitative agreement between the observed pre-martensitic anomalies, in contrast to titanium nickelide, the softening of the elastic constants is realized mainly by the Ziner channel C′ on the shear system {110}<110> (Figure 4a). Their changes are essentially anisotropic (A = C_44_/C′ >> 1), which correlates with the possibility of shuffling shear displacements of atoms and a noticeable softening of the phonon modes of the TA_2_ branch [19,29,30,31,32]. For example, for the copper-based alloys we have A = 12 − 13, and for the ferromagnetic intermetallic compound Ni_2_MnGa, A = 23 [33,34]. It is important that in the Ti–Ni–Cu-based alloys the parameter of anisotropy increases also almost twice in comparison with that of a binary TiNi compound (Figure 4b).

In recent years, a number of papers have widely discussed the important role of the weighted average number of valence electrons per atom e_v_/a and their concentration c_v_ = e_v_/e_t_ (where e_t_ = Z is the total number of electrons) in the temperature–concentration behavior of temperatures of TMTs as the factors responsible not only for elastic properties, but also for the overall stability of the atomic crystal lattice of the austenite in relation to the TMTs [35,36,37,38]. It was suggested that variation in the electronic band structure must be explicitly considered [38]. Valence electrons govern bond formation and thus the elastic properties [35,36]. In particular, on a large number of different doped alloys (more than 200 different alloys with TMTs) it was found that their martensite start temperatures M_s_ decrease almost linearly with increasing c_v_, although the data show large scatter for different alloys [37,38]. Conceptionally, c_v_ mixes parameters such as the valence electron number and atomic size [38]. The martensite temperatures M_s_, M_f_, A_s_, A_f_ and their dependence on alloy composition are of the utmost importance in shape memory technology [1,2,3,4,5,6,7,8,9,10,21,28,32,38].

As is known, TMTs and related SMEs in the binary and low-doped virtually equiatomic Ti–Ni alloys are realized in the operating temperatures below 373 K, limiting or excluding some useful practical applications of them. First, this is due to the “wedging” of the homogeneity region of existence of the phase B2 in the phase equilibrium diagram to the stoichiometric composition of Ti_50_Ni_50_ below 900 K, which leads to decomposition of non-stoichiometric Ti–Ni-alloys during heat treatment.

Second, the changes of the chemical composition of binary Ti–Ni alloys with the SMEs are limited between 50.0 and 51.2 at.% Ni due to the strong concentration dependence of the start (M_s_, A_s_) and finish (M_f_, A_f_) temperatures of the forward and reverse TMTs. They are almost linearly reduced by more than 100 K with gradual addition to 1 at.% Ni [1,2,3,4,5,6,32,38]. It is also well known that small impurities of O and C lead to a decrease in the TMTs temperatures in TiNi-based alloys and worsen their physical and mechanical properties because of forming Ti_4_Ni_2_O_x_ oxides and TiC carbides [1,2,3,4,5,6]. 

The strong influence of the concentration fluctuation of the basic components on the properties is not typical of commercial structural and functional materials. On the contrary, these materials must have a weaker concentration dependence of their exploitation characteristics and a relatively large composition tolerance, for example, in steels of no less than of 0.5–1.0 at.% content variations, which, as a rule, does not require constant precision chemical control and is technologically quite simply achieved [38,39]. Therefore, at least in the last two decades, active investigations have been conducted to find ternary and multicomponent compositions of alloys with SMEs as candidates capable of expanding and surpassing TiNi-based alloys’ unique capabilities. Doping by the various chemical elements results in different influence on the TMTs and SMEs in B2–TiNi alloys. Their solubility varies in wide limits and depends on the type of replacement Ti- or Ni-sublattice in the initial B2 superstructure into which the substitution is occurred in the binary equiatomic alloy TiNi—by the sections Ni_50_Ti_50−x_Me_x_, Ti_50_Ni_50−y_Me_y_ or by the other way [4,6,21,31,32,38,40,41]. The analogous data were received for ordered Cu–Al–Ni and other Cu-based alloys [1,2,3,4,5,6].

Ternary alloys of Heusler, of L2_1_ type, based on Ni_2_MnGa represent another important class of low-modulus ferromagnetic alloys with TMTs, in which the SMEs and PE can be induced by a magnetic field along with thermal and mechanical action [33,34,42]. For these brittle alloys, the problem of complex doping and optimization of their microstructure to provide the necessary TMT temperatures and the required physical and mechanical properties is even more necessary.

The behavior of key SME characteristics of alloys with concentration deviation from the quasibinarity is practice-important. In commercial synthesis of multicomponent materials it is difficult to obtain large-volume ingots with high-precision chemical composition and at the same time to avoid its uncontrolled deviation. Of course, it is necessary to take into account this fact in the design and development of new alloys, for example, because of a possible decomposition that determines significant changes in their physical and mechanical properties and, especially, ductility, in nonstoichiometric TiNi-based alloys [32,38,42,43,44,45,46]. In the alloys with high-temperature SMEs, the effect of homogeneous and/or heterogeneous decomposition is also responsible for changes in the properties during their exploitation [47,48,49,50]. Moreover, along with the doping and homogenizing treatment of the initial alloys, for the effective controlling of the properties one usually tends to employ—in the capacity of the economic alternative—the modern progressive methods that could radically change the alloys’ microstructural state (grain size, substructure and inclusions of excess phases). At the same time, on the one hand, various mechanisms of hardening (solid solution, deformation, aging or grain refinement) can be additionally implemented, and, on the other hand, some negative effects of alloys in the initial cast state can be weakened or modified.

The aim of the present work is the comparative systematic study of the structural and phase transformations, physical and mechanical properties of binary and ternary alloys of quasi-binary compositions with SMEs, based on the B2–TiNi, L2_1_–Ni_2_MnGa and D0_3_–CuAlNi compound systems, depending on high-doping and employment of extreme thermal and mechanical treatments with the use of rapid quenching from the melt (RQM) or severe plastic deformation (SPD).

## 2. Materials and Methods

The initial components of high-purity better than 99.95 wt.% were chosen for synthesis of the alloys. The alloys were obtained by the arc melting method in a helium atmosphere. Table 1 and Table 2 represent the studied binary and ternary alloys based on titanium nickelide, and Table 3 and Table 4 ternary alloys based on Ni_2_MnGa and Cu–Al–Ni (concentrations are given in at.%). All the alloys were re-melted three times to ensure a uniform chemical composition of the ingots. In our homogenization procedure we used hot pressing or forging of ingots by 5–10% at 1173–1273 K, subsequent annealing in vacuumed quartz capsules at 1223 K for 24 h, and quenching in water. After this treatment, generally homogeneous fine-grained (FG) microstructures (with grain sizes less than 100 µm) were obtained in the Ti–Ni-based alloys, in distinction from large-grained (LG) brittle homogenized as-cast Ni–Mn–Ga-based alloys. SPD was provided by equal-channel angular pressing (ECAP), high-pressure torsion (HPT) and multi-pass extrusion into a wire or strip. RQM ribbons with a thickness of 30 µm and a width of 1.5 mm were manufactured by quenching from the melt on a rotating copper disc, providing the cooling rate of 10^5^–10^6^ K/s. The structural-phase composition was determined by X-ray diffraction (XRD) in monochromatic radiation CuKα. Micro-structural and micro-chemical analysis was performed using the transmission and scanning electron microscopes (TEM) Tecnai G^2^ 30 and (SEM) Quanta 200 Pegasus equipped with an analytical energy dispersive X-ray spectrometry (EDS) system. TEM studies included direct observation and in-situ experiments on heating or cooling. In SEM studies, EBSD and fractography of samples subjected to tensile testing were also used. TMTs temperatures (M_s_, M_f_, A_s_ and A_f_) were measured by temperature dependences of the electrical resistance and magnetic susceptibility. Mechanical properties (σ_m_, σ_y_, σ_u_, σ_r_, δ, ε and Ψ) were determined on Instron-type tensile devices at room temperature. SMEs were fixed by heating the samples to temperatures above A_f_, restoring their shape after bending by means of special device in the martensitic state.

## 3. Results and Discussion

### 3.1. Temperature and Concentration Dependences of TMT Temperatures

The obvious necessary conditions for the synthesis of the alloys with SMEs based on titanium nickelide are the preservation of the solid solution state in the high-doped replacement-undergoing sublattice and in general in the B2 superstructure, as well as its ability to TMTs. The ternary alloys selected for systematic study, the matrix of which was as close as possible to quasi-binary systems in chemical composition (taking into account the possible precipitation of the TiC and Ti_4_Ni_2_O_x_ phases, increasing in general by 0.2–0.5 at.% concentration of nickel in the austenitic B2 matrix) corresponded to such conditions. Table 1 and Table 2 show the TMTs temperatures M_s_, M_f_, A_s_ and A_f_ of the binary and ternary quasi-binary alloys of titanium nickelide considered in the present paper (concentrations are given in at.%), average numbers of valence electrons per atom (e_v_/a) and their concentration c_v_ = e_v_/z. The alloys for the study were selected so that the nominal content of their B2 austenite did not differ from the real one by more than ±0.1 at.%. From the Table 1 and Table 2 it followed that in dependence of the doping of the third component the quasi-binary alloys of titanium nickelide underwent TMTs and, respectively, SMEs in a wide range of temperatures, from cryogenic to 1000 K, and all these determined their unique practical potential.

According to XRD, TEM and SEM, it was found that all the alloys were in a single-phase austenite state (in the presence of a certain small amount of carbides and titanium oxides, undetectable by XRD). Thus, high-doped alloys of studied compositions were solid solutions and it could be assumed that the filling of the alloying addition occurred purposefully in one of replacement-experiencing Ti or Ni sublattices of the B2 superstructure TiNi. In accordance with data from Table 1 and Table 2 we replaced Ni by the Al, Mn, Fe, Co, Cu, Pd, Pt and Au or, on the contrary, Ti by the Al, Mn, Zr and Hf. The results show a pronounced composition dependence of not only TMTs temperatures, but also the number of the valence electrons and their concentration. Adding chemical elements that in the periodic table are right below Ni (period VIIIB: Pd or Pt) or Ti (period IVB: Zr and Hf) resulted in the strong increase of M_s_, M_f_, A_f_ and A_s_. On the contrary, the addition of chemical elements that in the periodical table are located between Ti and Ni (Cr [38], V [38], Mn, Fe, Co) resulted in a decrease of TMTs temperatures.

The presence of a two-stage TMT B2 ↔ R ↔ B19′ with a sharp decrease of the temperatures for the last transformation R ↔ B19′ was common to the alloys, and is given in one Table 1. The coincidence of the trends of their changes with changes of e_v_/a and c_v_, as well as in the high-doped alloys in a solid solution state on the base of L2_1_–Ni–Mn–Ga and D0_3_–Cu–Al–Ni (Table 3 and Table 4, Figure 5), was unusual and contradicting to the conclusions of the valence electron concentration hypothesis proposed by Zarinejad and Liu [35,36,37]. The only correlation in the changes of the martensite temperatures and c_v_ in the TiNi-based alloys doped by Mn, Cr and V (replacing Ti) was in accordance with the data [37] (see Table 1 and Table 5). We used data from [38] on high-purity titanium nickelide alloys doped with Cr, V or Cu, and from [43] on conventional alloys, with Cu for comparison.

The behavior of the TiNi alloys doped with Cu, Pd, Pt and Au (replacing Ni) or Zr and Hf (replacing Ti; see Table 2) was even more unusual. In the case of Cu-, Zr- and Hf-doped alloys undergoing TMTs B2 ↔ B19′ the opposite changes of martensite temperatures and electron concentrations did occur. However, in the alloys moderate-doped with Pd, Pt and Au the first B2 ↔ B19′ TMTs were distinguished by coincidence of its changes. The opposite changes of martensite temperatures occurred during the second TMTs B2 ↔ B19. Adding Cu up to 15 at.% decreased all martensite temperatures of TMTs B2 ↔ B19′, but slightly increased the temperatures of the TMTs B2↔B19 (Table 2). Starting from Cu 25 at.%, the Ti_50_Ni_50−x_Cu_x_ alloys undergo eutectoid decomposition with the formation of equiaxial and lamellar dispersed particles of the B11-TiCu phase [44,45]. 

It should also be noted that when doping with the third component, it is performed from common Ti and Ni periods IVB and VIIIB of the periodic table, that is, e_v_/a = 7, the TMTs temperatures increased as much as possible. However, c_v_ from the value 0.28 corresponding to Ti_50_Ni_50_ could decrease in different ways correlated with the position of the element in the large periods of the periodic table and, consequently, with the number Z of the chemical element. Discussing the quantitative dependence of the martensite temperatures, e_v_/a and c_v_, we could still point to their significant numerical differences for various alloys with similar temperatures of TMTs. For example, for the B2–NiMnGa-based and D0_3_–CuAlNi-based alloys they were much larger than for alloys based on the B2-compound TiNi (Table 3 and Table 4).

The influence of the alloy decomposition at temperatures of TMTs is a known important fact noted already in the discussion on the B2 alloys Ti–Ni–Cu [44,45]. At the same time, it is obvious that not only the mechanical properties of the alloys were changed, but also the temperature characteristics of the TMTs decreased. This is indicated by the data of Table 2 on the change in the temperature–concentration dependence of TMTs at a concentration of Cu more than 25 at.%, equal to the solubility limit of Cu in B2–TiNi. Such behavior of alloys was also found in varying the content of not only Cu, but also Ti and Ni in the alloys of this system, which led to decomposition [44,45]. Significant influence of small impurities O and C on the TMTs, structure and properties of titanium nickelide binary alloys was already noted. Their great role in Ti–Ni–Cu alloys with TMTs was evidenced from the data of Table 5. Thus, alloys of Ti–Ni–Cu of the same chemical composition deteriorated more by C and O having lower TMT temperatures, which could be explained only by their solid solution effect. The negative role of inclusions of oxides and carbides was great in the ductile behavior of alloys, increasing their brittleness.

### 3.2. Crystal Structure and Morphological Regularities of Alloys with TMTs

Table 6 and Table 7 show the effect of doping on the phase composition, on the lattice parameters of the austenitic and martensitic phases and on the volume TMTs effect, ΔV/V, with comparison to the known literature data [46,47,48]. The given results demonstrate regular concentration dependences of the martensite structural type, parameters of their unit cell and rather small values |V/V| ≤ 1%, typical of TMTs.

According to SEM and TEM investigations, the packet morphology of pairwise-twinned martensitic crystals with coherent boundaries was typical of the martensitic microstructure of all studied quasi-binary fine-grained (FG) alloys (Figure 6). As is seen from Figure 6b,c, the pressure-induced martensitic crystals in low-module alloys would be oriented in each grain in the preferred crystallographic directions, demonstrating the microstructural mechanism of the pseudoelasticity effect due to macro deformation.

The flat coherent boundaries of crystals of B19′ martensite of different orientations within the packages occur, as a rule, along the planes (011), as well as (111) and (113)_B19′_ [40,41,49,50,51,52,53,54]. The packet character of the morphology of the B19′ crystals pairwise-twinned by I-type (011)_B19′_ was clearly visible on TEM images at such grain orientations of twinned B19’ crystals when the zone axes (z.a.) of reflecting planes was [100]_B19′_ (Figure 7a and Figure 8a). Note also that thin secondary nanotwins were often clearly visualized within the wider initial twinned crystals of the packets.

Figure 7c, Figure 8b,c and Figure 9 present another typical example of TEM images and SAEDs of twinned martensite of packet morphology of twinned martensite, when zone axes were [110]_B19′_. In this case, parallel-plate or wedge-shaped crystals of the initial orientation forming the package took turns with crystals of twins of I-type (011)_B19′_ and II-type <011>_B19′_. There were martensite packets with thin-twinned crystals or with wider plates, internally thin-twinned already on secondary systems of twinning, oriented at an angle to the boundaries of primary crystals (Figure 7c and Figure 8a,c). There were nanotwins of I-type on (111) (Figure 9) and on (001) B19′ (Figure 7c, Figure 8b,d and Figure 9b). In this case, in SAED (Figure 7d, Figure 8d and Figure 9d) sharp streaks correspondent to them in the directions of the reciprocal lattice [111]*_B19′_ and [001]*_B19′_, respectively, were observed. It was found that the orientation ratio (o.r.) of B2 austenite and B19′martensite crystals was close to Bain one:(100)_B2_||(100)_B19′_; [011]_B2_||[010]_B19′_; [011]_B2_||[001]_B19′_(1)

As is known, in the B19′ martensite substructure of both binary and ternary titanium nickelide alloys, the twin mode of II type <011> plays an important role, responsible for the homogeneous shear deformation necessary to retain the macroscopic invariance of the habit plane at the TMT B2 → B19′ in accordance with the concepts of phenomenological crystallographic theory [47,49,51,52]. In B19′ martensite alloys with Cu, Zr, Hf, along with twins of II-type <011>_B19′_ a large number of twins of I-type (011) and (111)_B19′_, as well as composite twins on (001)_B19′_ were present. Obviously, the presence of all of these mentioned nanotwins of the I and II types provided both geometrically necessary shear with invariant lattice in the TMTs B2 → B19′ and pseudo-elastic volume and planar accommodation of the martensitic crystals due to the packet pair-twinned morphology and, if necessary, secondary nanotwinning in these low-modulus alloys of titanium nickelide [49,51,52]. This can explain the variety of observed variants of coherent boundaries and twinning of martensite crystals. It is assumed that pseudo-elastic composite nanotwins on (001)_B19′_ are often observed both in the depleted and enriched TiNi-based alloys, which are not geometrically necessary, and also provide for additional elastic-plastic accommodation of the lattice of B19′ martensite [40,41,48,49].

B19 martensite formed in titanium nickelide alloys doped with Cu, Pd, Pt and Au also had a predominantly packet-pyramidal morphology of pairwise-twinned plate crystals (Figure 10a–d). Secondary nanotwins appeared under subsequent cooling in B19 crystals or at B19 → B19′ TMTs (Figure 10e,f). Similar results were obtained on B19 martensite in alloys doped with Pd, Pt and Au when the temperature was reduced down to room temperature. The orientation ratio of the lattices of the B2 austenite and the B19 martensite was similar to the o.r. (1).

A fine structure and morphology of the packet 2 M and long-period 10 M and 14 M martensites in large-grained (LG), FG, and ultra-fine-grained (UFG) alloys based on Ni_2_MnGa (Figure 11 and Figure 12) and 2 H and 18 R long-period of martensites in alloys based on Cu–Al–Ni were qualitatively similar [41].

An important feature of titanium nickelide doped by copper instead of nickel in concentrations exceeding 23–25 at.%, hafnium and zirconium instead of titanium exceeding 10–12 at.% is their ability to synthesize the alloys in the amorphous state by spinning from the melt [55,56,57,58,59,60,61,62,63,64,65,66,67,68,69]. As a result of the subsequent optimal heat treatment, it was possible to provide the formation of predominantly single-packet martensite with high-temperature SMEs, a high-strength and ductile UFG structure (Figure 13). 

Another effective external influence for the radical grain-size refinement of up to amorphization is provided by various methods of plastic deformation including HPT, ECAP, multi-pass cold rolling, cold drawing, local canning rolling or compression, plain strain compression, screw extrusion, and others [70,71,72,73,74,75,76,77,78,79,80,81,82,83,84,85,86,87,88,89,90,91]. Figure 14 illustrates the main types of microstructures obtained in the HPT and ECAP. 

### 3.3. Mechanical Properties of Fine- and Ultrafine-Grained Alloys

Table 4 presents the tensile mechanical properties of the quenched Cu–Al–Ni-based alloys with different grain sizes, Table 8—FG titanium nickelide alloys of normal purity of O and C, including the dopants Cu, Fe, Co and Pd, at room temperature, and Table 9 for comparison—binary FG alloys of high purity in C and O. At close values of the average grain size (50–70 µm) the alloy Ti_49.4_Ni_50.6_ had almost twice the larger relative elongation (δ = 75%), than the best in Table 8 in the properties of the Ti_50_Ni_50_ alloy (δ = 40%). It is seen that the doping of Cu, Fe, Co and Pd of alloys led to a decrease in the relative elongation, although their strength characteristics and SMEs remained attractive. The strongest concentration changes in the mechanical properties and grain size, accompanied by sharp embrittlement, were demonstrated by copper alloys (Table 4). 

A noticeable improvement in the strength and ductile properties of titanium nickelide was achieved in the formation of the UFG structure that was formed via in advanced deformation thermal technologies using SPD (ECAP, HPT, multi-pass rolling and drawing into a strip, rod or wire) of initial large-size samples of Ti–Ni alloys (Table 9) [51,52,53,54,55,56,57,58,59,60,61,62]. 

RQM methods can be most useful for creating thin UFG tapes with SMEs as sensors or actuators in miniature MEMS devices [55]. Their usage after additional subsequent annealing allows obtaining long homogeneous tapes with the UFG structure, attractive in mechanical properties and parameters of the SMEs. A complex of high mechanical properties of the rapidly quenched UFG alloy doped with copper (σ_u_, 850–1550 MPa; σ_y_, 620–1200 MPa; σ_m_, 100–50 MPa; δ, 9–12%; reactive stress σ_r_ = σ_y_−σ_m_, 620–1110 MPa and a reversible deformation of the ε, 3–5%; Table 10) was received and a new method of obtaining high strength UFG SMEs alloys in the form of thin ribbons was proposed, based on the technology of spinning from the melt of the non-stoichiometric alloys Ti_50+x_Ni_25−x_Cu_25_ (x ≤ ±1 at.%) and Ti_50+y_Ni_25_Cu_25−y_ (y ≤ ±1 at.%) [44,45]. Similar structural changes needed for increasing strength and ductile properties were also found on Ni–Ti–Zr, Ni–Ti–Hf and (Ni, Cu)_50_(Ti, Hf)_50_ alloys [56,57,58,59,60,61,62,63,64]. Thus, it was found that the creation of the FG and UFG structure could significantly improve or (in the case of strong strengthening due to extreme external influences) preserve the ductility of the alloys necessary for the implementation of SMEs.

In conclusion, the results of fractographic SEM studies of alloys with TMTs were studied. It was found that in the quenched FG alloys Ti_50_Ni_50_ and Ti_49.4_Ni_50.6_ the fracture had generally a viscous transgranular character (Figure 15a). The creation of the UFG structure did not change the type of fracture and the character of the destruction of these alloys. Many centers of localization of deformation with the appearance of small flat pits and, consequently, low ridges of separation, which was typical of the viscous mechanism of fracture with low energy, were observed on the surface of the fractures (for example, Figure 15b). The quenched large-grained (LG) TiNi-based alloys doped with Cu, Zr and Hf underwent the quasi-brittle fracture. There took place classic brittle fracture with predominant plane cleavage along the grain boundaries in the Cu–Al–Ni-based alloys (for example, Figure 15c).

In our opinion, the increase of the elastic anisotropy in the high-doped alloys based on TiNi was the main reason for the strong dependence of their mechanical properties on the grain size. All other alloys with TMTs of Ni–Mn–Ga and Cu–Al–Ni systems also were characterized by the utmost high elastic anisotropy. It was because of the coherent accommodation of elastic stresses induced by the volume effect; they were localized at the grain boundaries, which ultimately becomes the general cause of intergranular brittleness of the alloys with large grain size, undergoing TMTs.

## 4. Summary and Conclusions

In the present work, ternary close to quasi-binary alloys based on the Ti–Ni–X system (where X = Al, V, Cr, Mn, Fe, Co, Cu, Zr, Pd, Hf, Pt or Au) and, for comparison, alloys of two actual systems Ni–Mn–Ga and Cu–Al–Ni were selected to investigate the possibilities for the creation of strength and ductile materials with TMTs and related SMEs due to multicomponent alloying and extreme external treatments. TMT temperatures, the type and parameters of the martensitic phases, the volume effect of the phase transition, the average number of valence electrons (e_v_/a) and their concentration (c_v_) were determined. A detailed study of the fine structure and morphological features of martensite in large, fine and ultrafine-grained alloys (after their synthesis by rapid quenching from the melt or severe plastic deformation) was carried out. Tensile mechanical properties of a number of alloys were measured and mechanisms of their destruction were established. From the analysis of the obtained results, taking into account the literature data, the following conclusions could be made:
The small volume effect |ΔV/V| < 1%, typical of TMTs, as well as the presence of pre-martensitic softening of elastic constants and the formation of a special heterogeneous pre-martensitic state was common for all studied alloys with a sufficiently noticeable temperature–concentration changes of crystal-structure parameters of austenite and martensitic phases, structural types of martensite and their metastable long-period variants.All studied alloys with TMTs combined the similarity of the pair-twinned packet morphology and microstructural hierarchy, proximity to Bain ratio, the action of several geometrically necessary twinning systems of martensite and the presence of coherent variants of their boundaries and subboundaries. In the condition of low elastic constants this diversity of substructure elements in thermoelastic martensite provided pseudo-elastic planar and volume lattice accommodation, and in the case of external mechanical influences, their mobile favorable reorientation and the effect of pseudoelasticity.The preservation of their single-phase high temperature austenitic state capable of TMTs played an important role in the multicomponent doping of these alloys. The chemical elements Zr and Hf from the Ti-period IVB of the periodic table had increased solubility (up to 25–30 at.%) replacing Ti in the quasi-binary alloys NiTi–NiMe. The elements Fe, Co, Pd and Pt from the Ni-period VIIIB had unlimited solubility replacing Ni in the quasi-binary alloys TiNi–TiMe. The solubility of the remained metals in the B2–TiNi compound was relatively low (up to 10 at.%), except for the unlimited solubility of Au and high (up to 25 at.%) one of Cu. It is essential that the high doping by Zr, Pd, Hf, Pt and Au provided a noticeable increase in the temperatures of TMTs, extending the temperature range of the implementation of TMTs above 373 K up to 1000 K. While Fe, Co and a number of other transition metals V, Cr and Mn, located between the elements Ti and Ni, caused them to noticeably decline below room temperature. When comparing the concentration changes of TMTs temperatures, average concentrations of valence electrons (c_v_) and the number of valence electrons (e_v_/a) for the alloys of the studied doping systems, a wide variety of them were established, which in general did not allow us to correctly determine their correlation. For example, for a number of alloys with high-temperature TMTs high-doped with Zr and Hf (replacing Ti) or Cu, Pd, Pt and Au (replacing Ni), the strong increase in TMTs temperatures was indeed consistent with the opposite decrease in the c_v_ value. There was also the divergence between these factors and doping with V, Cr and Mn. On the contrary, there was a coinciding change in the other TiNi-based alloys moderately doped by the elements Al, Fe, Co, Cu, Pd, Pt and Au replacing Ni. In L2_1_–Ni–Mn–Ga and D0_3_–Cu–Al–Ni alloys, a strong increase in TMTs temperatures was also accompanied by a noticeable increase in c_v_ and e_v_/a. The strong dependence on the grain size characterized the mechanical behavior of the studied alloys during tensile tests, especially for the relative elongation. The strength and the ductile properties of the ternary alloys decreased and the fracture became more brittle intercrystalline, in contrast to the viscous transcrystalline character in the binary TiNi alloy.In our opinion, the increase of the elastic anisotropy in high-doped alloys based on TiNi was the main reason for the strong dependence of their mechanical properties on the grain size. All other alloys with TMTs of Ni–Mn–Ga and Cu–Al–Ni systems also were characterized by the utmost high elastic anisotropy. It is because of the coherent accommodation of elastic stresses induced by the volume effect, they were localized at the grain boundaries, which ultimately becomes the general cause of intergranular brittleness of the alloys with large grain size undergoing TMTs.The strength and ductility of alloys with TMTs required for the realization of SMEs could be achieved by doping, which would provide a decrease in the elastic anisotropy parameter, and on the other hand, due to the formation of a fine-grained structure. Important especially for alloys with high temperature SMEs was the presence of homogeneous and heterogeneous decomposition with a strong strengthening effect that one should avoid using the quasi-binary precision doping.


In conclusion, based on the obtained experimental results we suggest dividing various shape memory alloys to two groups. The first group shows an extremely strong dependence of martensite temperatures on the alloy composition. This is, for example, observed for high-doped ternary Ti–Ni–Pd, Ti–Ni–Pt, Ti–Ni–Au, Ni–Ti–Zr and Ni–Ti–Hf alloys with high-temperature TMTs. The behavior of these alloys was well predicted by the valence electron concept proposed by Zarinejad and Liu [35,36,37] and a gradual change in the enthalpy of transformation [38]. In these alloys the B2 phase cannot be stabilized by the formation of antisite defects, because the alloy additions also stabilize martensitic phases B19 or B19′ [38]. On the contrary, the second group of Ni–Ti, Ni–Mn–Ga and Cu-based intermetallic alloys showed opposite dependence of martensite temperatures on the alloy composition and electron concentration when the trends of their changes coincided (decreased or increased together). This was observed for the binary TiNi and low and moderately doped ternary Ti–Ni–Al, Ti–Ni–Fe, Ti–Ni–Co, Ti–Ni–Cu, Ti–Ni–Pd, Ti–Ni–Pt and Ti–Ni–Au. The compositional dependence of martensite temperatures in these alloys can be rationalized on the basis of a strong stabilization of B2-austenite through the formation of antisite defects [38].

## Figures and Tables

**Figure 1 materials-12-02616-f001:**
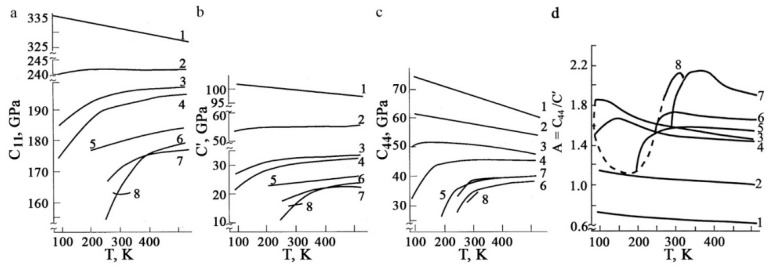
Temperature–concentration dependences of elastic constants: (**a**) C_11_, (**b**) C′, (**c**) C_44_ and (**d**) elastic anisotropy parameter A = C_44_/C′ of the single-crystalline alloys: 1—Ti_50_Fe_50_, 2—Ti_50_Ni_25_Fe_25_, 3—Ti_50_Ni_35_Fe_15_, 4—Ti_50_Ni_40_Fe_10_, 5—Ti_50_Ni_45_Fe_5_, 6—Ti_50_Ni_48_Fe_2_, 7—Ti_50_Ni_50_ and 8—Ti_49_Ni_51_ [11]. (Reproduced with permission from Khachin V.N.; Muslov S.A.; Pushin V.G.; Chumlyakov Y.I., 1987, DAN SSSR).

**Figure 2 materials-12-02616-f002:**
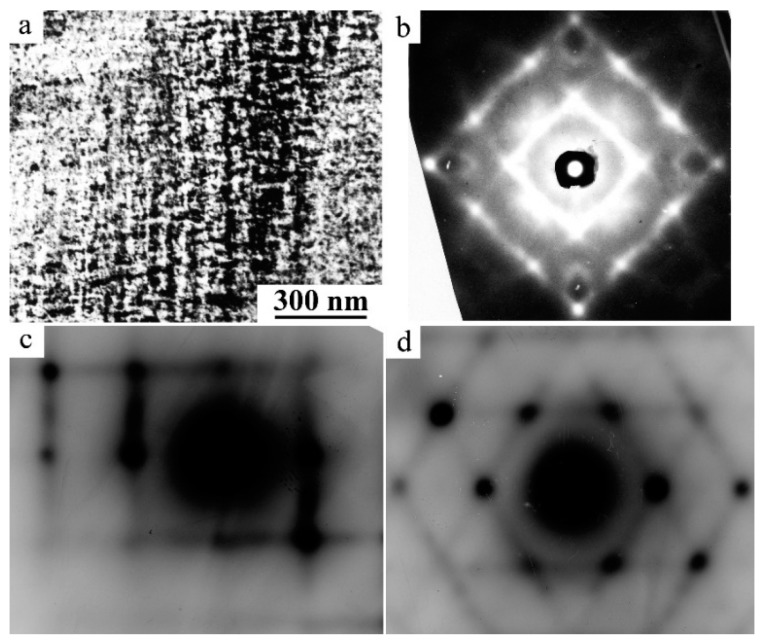
Typical image of (**a**) tweed TEM contrast and (**b**–**d**) diffuse scattering of single-crystalline alloys of titanium nickelide; (**b**) X-rays and (**c**,**d**) selected area electron diffraction (SAED) [4,6].

**Figure 3 materials-12-02616-f003:**
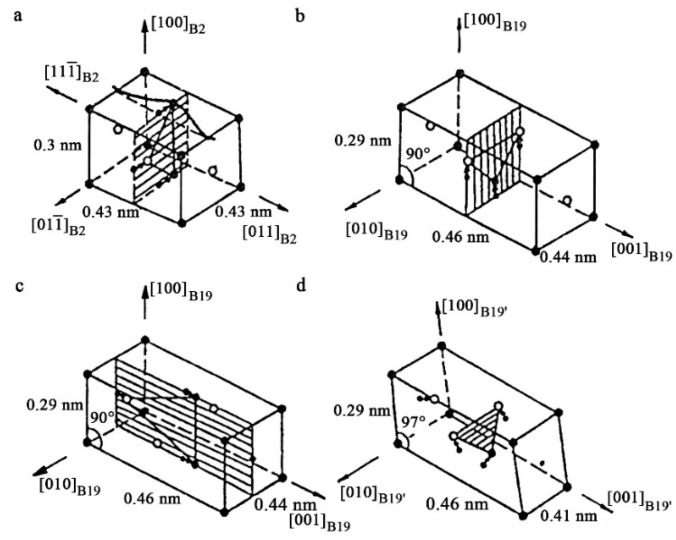
Unit lattice cells of the (**a**) B2-, (**b**,**c**) B19- and (**d**) B19′ phases in titanium nickelide alloys, their dimension-orientational ratios and schemes of rearrangements determined by shuffling (of type {011}<100> and {011}<011>) atomic displacements (shear basal planes {011}_B2_ shaded) [17] (Copyright, 1989, Physics of Metals and Metallography).

**Figure 4 materials-12-02616-f004:**
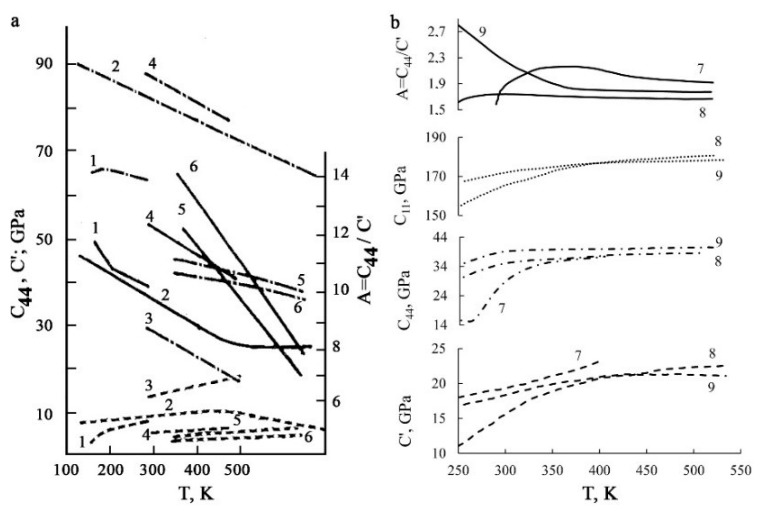
Temperature–concentration dependences of single-crystalline elastic constants C_11_ (dot line), C′ (dotted line), C_44_ (dash-dotted line) and the parameter of the elastic anisotropy A = C_44_/C′ (solid line); (**a**) BCC alloys: 1—Au-Zn-Cu, 2—Cu-Zn, 3—NiAl, 4—Cu-Al-Zn, 5—Au-50 at. % Cd and 6—Au-47.5 at. % Cd [19] and (**b**) 7—Ti_49_Ni_51_, 8—Ti_50_Ni_48_Fe_2_ and 9—Ti_50_Ni_38_Cu_10_Fe_2_ [12,13] (Copyright, 1987, Izvestia VUZov).

**Figure 5 materials-12-02616-f005:**
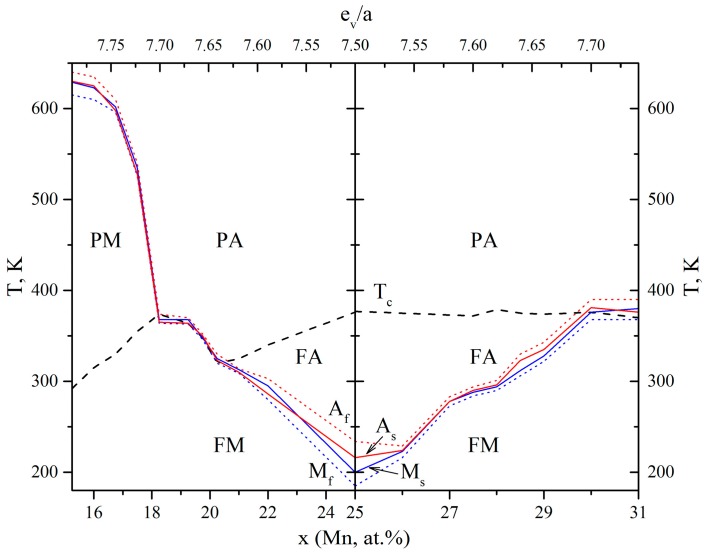
The temperature-concentration diagram of magnetic and martensitic transformations of Ni_50+x_Mn_x_Ga_25_ (left of x = 25) and Ni_50_Mn_x_Ga_50−x_-based alloys (right). The phase fields of paramagnetic (P) and ferromagnetic (F) austenite (A) and martensite (M) are indicated.

**Figure 6 materials-12-02616-f006:**
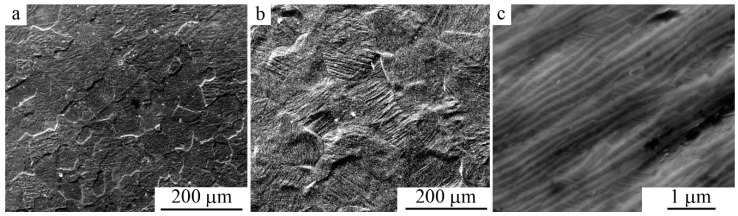
SEM images of the microstructure of Ti_49.4_Ni_50.6_ alloy: (**a**) In the austenitic state and (**b**,**c**) in the single-packet martensite state after treatment with a pressure of 6 GPa.

**Figure 7 materials-12-02616-f007:**
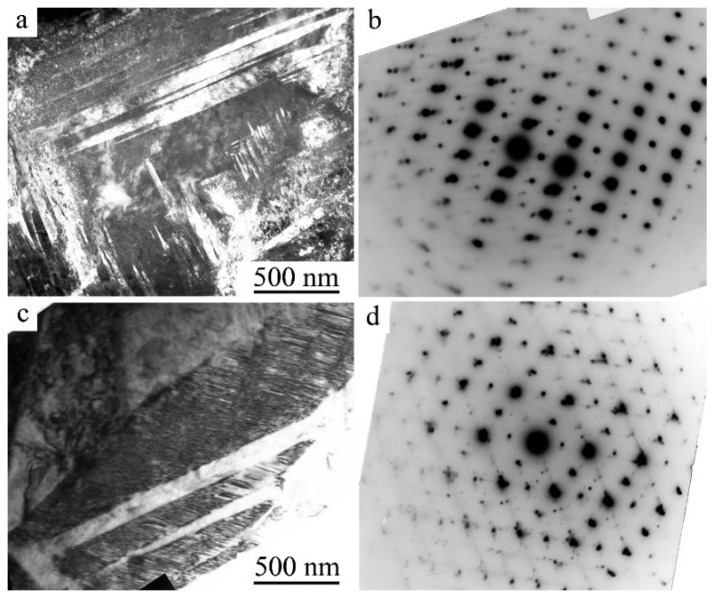
TEM images of the microstructure of B19′ martensite and the corresponding SAED (**a**,**b**) of alloys Ti_50_Ni_45_Cu_5_ at 250 K and (**c**,**d**) Ti_50_Ni_40_Cu_10_ at 120 K.

**Figure 8 materials-12-02616-f008:**
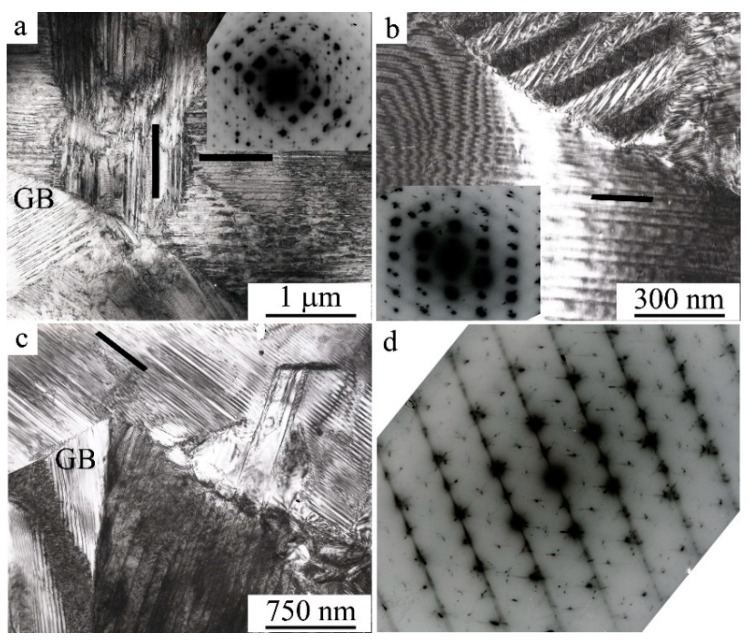
TEM images of the microstructure and the corresponding SAED of alloys Ni_49.5_Ti_50.5−x_Zr_x_: (**a**) x = 3 at.%, (**b**) x = 10 at.% and (**c**,**d**) x = 15 at.%.

**Figure 9 materials-12-02616-f009:**
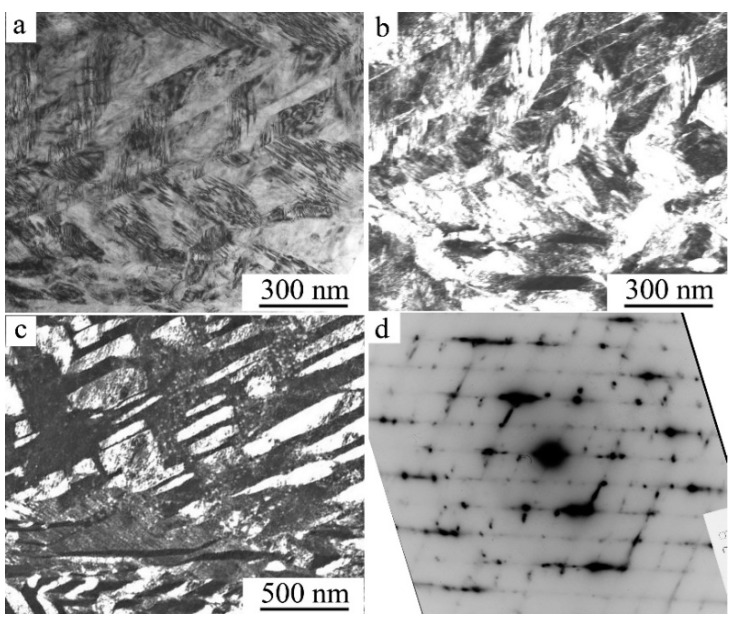
TEM images of the microstructure of the alloy Ni_50_Ti_32_Hf_18_ (**a**) bright field and (**b**,**c**) dark-field images and (**d**) correspondent SAED.

**Figure 10 materials-12-02616-f010:**
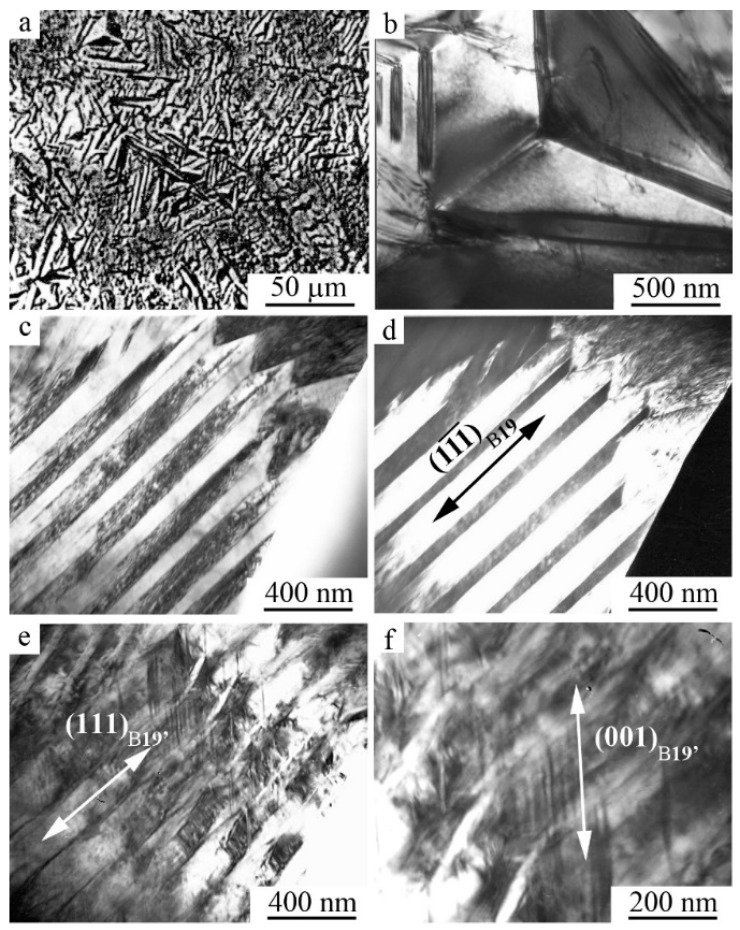
(**a**) SEM and (**b**–**f**) TEM microstructure images of Ti_50_Ni_40_Cu_10_ alloy obtained (**a**–**d**) at room temperature and (**e**,**f**) at 120 K.

**Figure 11 materials-12-02616-f011:**
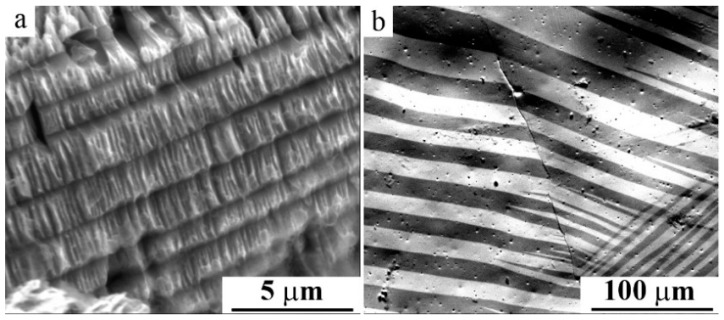
SEM images of (**a**) the packet twinned microstructure and (**b**) magnetic domain microstructure of the alloy Ni_50_M_n28.5_Ga_21.5_.

**Figure 12 materials-12-02616-f012:**
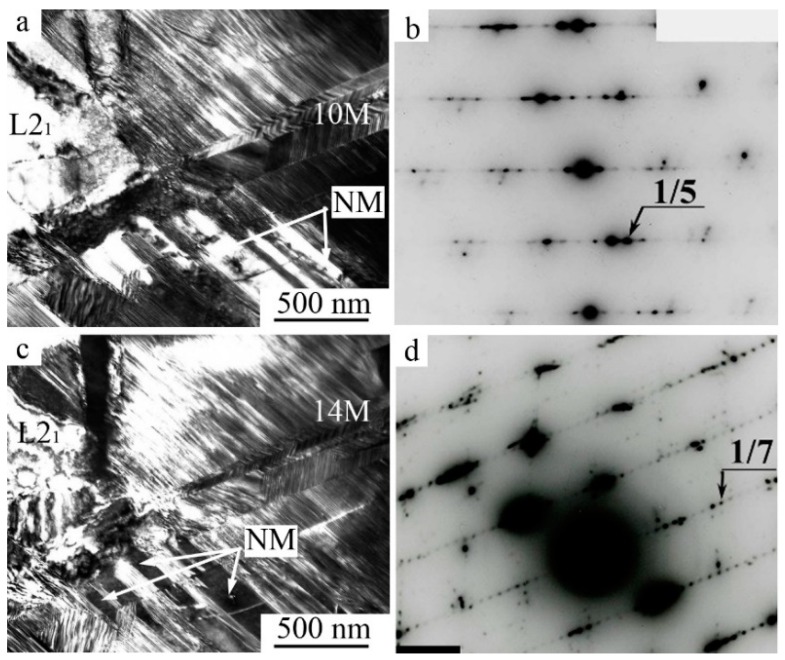
TEM images of the microstructure of the alloy Ni_54_Mn_21_Ga_25_ and related SAED (**a**,**b**) long-period 10 M martensite at room temperature and (**c**,**d**) long-period 14 M martensite at 130 K.

**Figure 13 materials-12-02616-f013:**
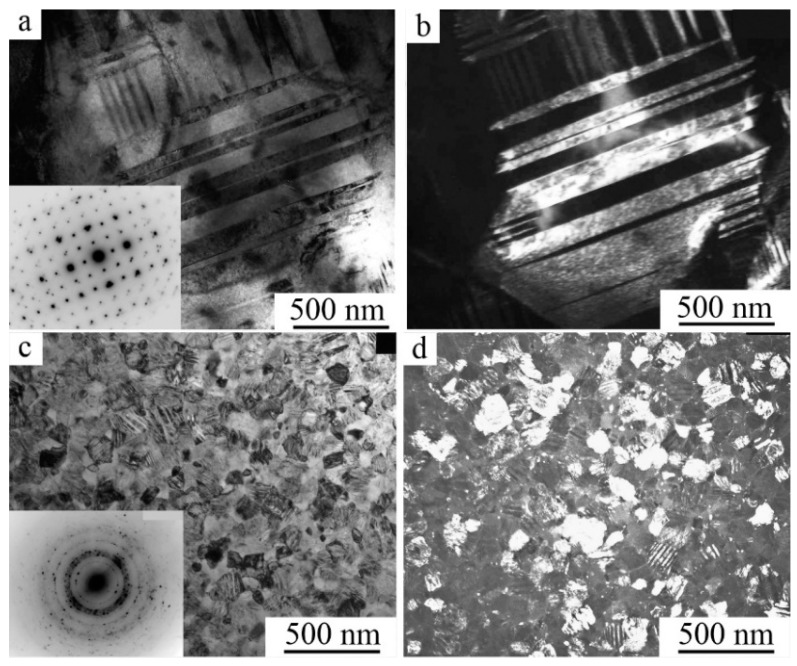
TEM images and corresponding SAEDs of rapid quenching from melt (RQM) alloys (**a**,**b**) Ti_50_Ni_25_Cu_25_ and (**c**,**d**) Ni_45_Ti_32_Hf_18_Cu_5_ obtained in an amorphous state and subjected to various crystallization annealing.

**Figure 14 materials-12-02616-f014:**
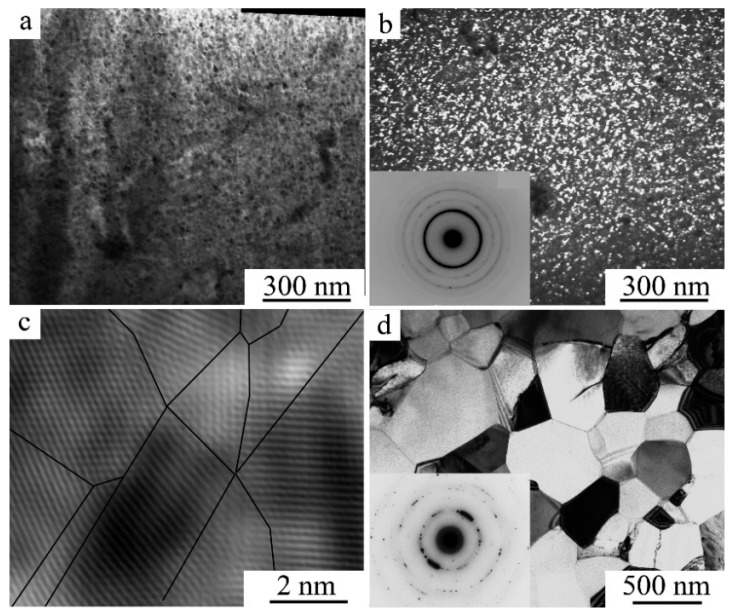
TEM images of alloy (**a**–**c**) Ti_49.4_Ni_50.6_ after high-pressure torsion (HPT), 5 revs, at 7 GPa, and subsequent annealing 523 K, 20 min and (**d**) Ti_49.8_Ni_50.2_ after step equal-channel angular pressing (ECAP) at 773 K + 723 K + 673 K, six passes.

**Figure 15 materials-12-02616-f015:**
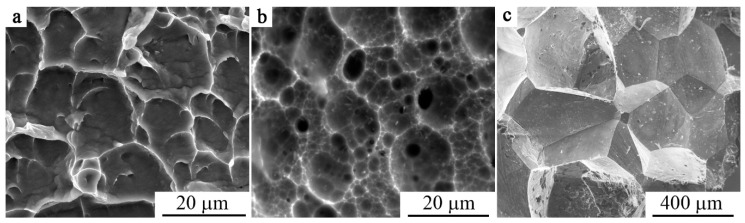
Fractography of (**a**,**b**) alloy Ti_49,4_Ni_50.6_ (**a**): After quenching, (**b**): HPT five turns at 6 GPa and subsequent annealing at 823 K and (**c**) quenched alloy Cu_69_Al_28_Ni_3_.

**Table 1 materials-12-02616-t001:** Chemical composition, temperatures of thermoelastic martensitic transformations (TMTs) B2 ↔ R ↔ B19′, average number of valence electrons per atom e_v_/a and their concentration c_v_.

Alloy, at.%	R				B19′				e_v_/a	c_v_
	M_s_′, K	M_f_′, K	A_s_′, K	A_f_′, K	M_s_, K	M_f_, K	A_s_, K	A_f_, K		
Ti_50_Ni_50_					343	319	353	373	7.00	0.280
Ti_50_Ni_49_Al_1_	319	305	301	323	264	245	289	309	6.93	0.279
Ti_50_Ni_48_Al_2_	295	280	276	301	200	178	227	253	6.86	0.278
Ti_50_Ni_47_Al_3_	262	247	242	270	130	105	161	188	6.79	0.277
Ti_50_Ni_46_Al_4_	190	179	171	198					6.72	0.275
Ti_50_Ni_45_Al_5_	139	131	112	142					6.69	0.274
Ni_50_Ti_49_Al_1_	321	301	305	324	266	247	288	309	6.99	0.281
Ni_50_Ti_48_Al_2_	300	272	277	303	199	162	222	251	6.98	0.281
Ni_50_Ti_48,5_Al_2,5_	282	253	256	285	109		179	211	6.98	0.282
Ni_50_Ti_47_Al_3_	260	228	232	265					6.97	0.282
Ni_50_Ti_46_Al_4_	173	134	147	184					6.96	0.282
Ti_50_Ni_49_Mn_1_	316	290	285	321	280	237	289	299	6.97	0.279
Ti_50_Ni_48_Mn_2_	295	263	259	301	208	164	230	259	6.94	0.278
Ti_50_Ni_47_Mn_3_	277	242	237	282	135	89	176	202	6.91	0.277
Ti_50_Ni_45_Mn_5_	237	191	189	243					6.85	0.276
Ni_50_Ti_49_Mn_1_	304	266	272	305	283	235	298	320	7.03	0.281
Ni_50_Ti_48,5_Mn_1,5_	296	259	266	301	247	191	265	290	7.05	0.281
Ni_50_Ti_48_Mn_2_	288	253	259	295	186	122	218	240	7.06	0.282
Ni_50_Ti_47_Mn_3_	253	216	223	262	78		112	140	7.09	0.283
Ni_50_Ti_46_Mn_4_	173	134	146	183					7.12	0.283
Ti_50_Ni_49_Fe_1_	307	294	298	313	278	250	289	305	6.98	0.279
Ti_50_Ni_48_Fe_2_	287	266	276	297	218	189	235	258	6.96	0.279
Ti_50_Ni_46_Fe_4_	264	238	252	280	155	111	186	210	6.92	0.278
Ti_50_Ni_45_Fe_5_	230	202	217	240					6.90	0.277
Ti_50_Ni_49_Co_1_	324	305	313	330	313	265	322	323	6.99	0.280
Ti_50_Ni_48_Co_2_	301	283	291	311	271	216	289	290	6.98	0.279
Ti_50_Ni_46_Co_4_	287	266	274	297	237	175	261	267	6.96	0.279
Ti_50_Ni_45_Co_5_	263	235	244	272	165	101	207	220	6.95	0.279
Ti_50_Ni_40_Co_10_	218	183	190	226					6.90	0.277
Ti_50_Ni_35_Co_15_	130	80	89	140					6.85	0.276

**Table 2 materials-12-02616-t002:** Chemical composition, temperatures of TMTs B2 ↔ B19 ↔ B19′, number of valence electrons per atom e_v_/a and their concentration c_v_.

Alloy, at.%	B19				B19′				e_v_/a	c_v_
	M_s_′, K	M_f_′, K	A_s_′, K	A_f_′, K	M_s_, K	M_f_, K	A_s_, K	A_f_, K		
Ti_50_Ni_45_Cu_5_					315	222	255	337	7.05	0.281
Ti_50_Ni_42,5_Cu_7,5_					305	209	229	318	7.08	0.282
Ti_50_Ni_40_Cu_10_	312	302	307	316	280	201	222	291	7.10	0.283
Ti_50_Ni_35_Cu_15_	332	317	322	337	200	181	186	207	7.15	0.284
Ti_50_Ni_33_Cu_17_	337	326	331	342					7.17	0.284
Ti_50_Ni_30_Cu_20_	337	327	332	345					7.20	0.285
Ti_50_Ni_25_Cu_25_	340	326	333	348					7.25	0.287
Ti_50_Ni_22_Cu_28_	335	323	328	342					7.28	0.288
Ti_50_Ni_18_Cu_32_	331	313	322	339					7.32	0.289
Ti_50_Ni_16_Cu_34_	321	307	313	330					7.34	0.290
Ti_50_Ni_48_Pd_2_					330	310	335	355	7.00	0.276
Ti_50_Ni_45_Pd_5_					320	300	330	350	7.00	0.27
Ti_50_Ni_40_Pd_10_	330	320	330	340	270	230	240	280	7.00	0.261
Ti_50_Ni_25_Pd_25_	450	410	430	470					7.00	0.237
Ti_50_Ni_5_Pd_45_	710	680	710	740					7.00	0.211
Ti_50_Ni_48_Pt_2_					305	280	305	330	7.00	0.270
Ti_50_Ni_42_Pt_8_	300	275	280	305	270	240	260	290	7.00	0.245
Ti_50_Ni_25_Pt_25_	720	670	720	770					7.00	0.194
Ti_50_Ni_47_Au_3_					315	280	320	350	7.03	0.265
Ti_50_Ni_45_Au_5_					360	305	320	375	7.05	0.256
Ti_50_Ni_38_Au_12_	365	340	355	380	300	255	250	310	7.12	0.229
Ti_50_Ni_25_Au_25_	425	365	400	460					7.25	0.192
Ti_50_Au_50_	880	850	870	900					7.50	0.149
Ni_50_Ti_47_Zr_3_					350	325	360	385	7.00	0.274
Ni_50_Ti_45_Zr_5_					350	325	370	395	7.00	0.27
Ni_50_Ti_40_Zr_10_					375	350	400	420	7.00	0.261
Ni_50_Ti_35_Zr_15_					455	425	475	500	7.00	0.253
Ni_50_Ti_32_Zr_18_					510	480	535	555	7.00	0.248
Ni_50_Ti_30_Zr_20_					555	520	570	600	7.00	0.245
Ni_50_Ti_38_Hf_12_					405	380	430	450	7.00	0.226
Ni_50_Ti_35_Hf_15_					450	430	480	500	7.00	0.215
Ni_50_Ti_32_Hf_18_					500	475	530	555	7.00	0.206
Ni_50_Ti_30_Hf_20_					520	495	564	585	7.00	0.200

The chemical composition of the basic elements varied within 0.1 at.% and O and C were within 0.07–0.1 wt.%.

**Table 3 materials-12-02616-t003:** Chemical composition, temperatures of TMTs L2_1_ ↔ 2M (14M), the number of valence electrons per atom e_v_/a and their concentration c_v_ of Ni–Mn–Ga-based alloys.

Alloy, at.%	M_s_, K	M_f_, K	A_s_, K	A_f_, K	e_v_/a	c_v_
Ni_50_Mn_50_	970	920	970	1020	8.50	0.321
Ni_50_Mn_48_Ga_2_	907	880	920	954	8.42	0.316
Ni_50_Mn_46_Ga_4_	860	830	860	897	8.34	0.312
Ni_50_Mn_44_Ga_6_	807	785	830	845	8.26	0.308
Ni_50_Mn_42_Ga_8_	755			790	8.18	0.303
Ni_50_Mn_40_Ga_10_	693			720	8.10	0.299
Ni_50_Mn_36_Ga_14_	546	525	540	557	7.94	0.290
Ni_50_Mn_34_Ga_16_	457	450	460	469	7.86	0.286
Ni_50_Mn_32_Ga_18_	378	375	375	390	7.78	0.282
Ni_50_Mn_31_Ga_19_	368	363	363	370	7.74	0.280
Ni_50_Mn_30_Ga_20_	362	356	370	374	7.70	0.278
Ni_50_Mn_29_Ga_21_	328	322	335	343	7.66	0.276
Ni_50_Mn_28.5_Ga_21.5_	312	306	323	330	7.64	0.275
Ni_50_Mn_28_Ga_22_	294	290	296	301	7.62	0.274
Ni_50_Mn_27.5_Ga_22.5_	288	284	290	294	7.60	0.273
Ni_50_Mn_27_Ga_23_	278	273	278	283	7.58	0.272
Ni_50_Mn_26_Ga_24_	223	216	224	229	7.54	0.270
Ni_50_Mn_25_Ga_25_	200	185	216	234	7.50	0.268
Ni_53_Mn_22_Ga_25_	295	280	286	303	7.59	0.270
Ni_54_Mn_21_Ga_25_	313	309	310	314	7.62	0.271
Ni_54.75_Mn_20.25_Ga_25_	325	320	323	330	7.64	0.272
Ni_55.25_Mn_19.75_Ga_25_	348	346	350	352	7.66	0.272
Ni_55.75_Mn_19.25_Ga_25_	368	363	364	370	7.67	0.272
Ni_56.75_Mn_18.25_Ga_25_	368	364	365	374	7.70	0.273
Ni_57.5_Mn_17.5_Ga_25_	535	526	528	540	7.73	0.274
Ni_58.25_Mn_16.75_Ga_25_	602	596	598	610	7.75	0.274
Ni_59_Mn_16_Ga_25_	623	610	625	635	7.77	0.275
Ni_60_Mn_15_Ga_25_	629	615	630	640	7.79	0.276

The chemical composition of the basic elements varied within 0.1 at.% and O and C were in the range of 0.1 wt.%.

**Table 4 materials-12-02616-t004:** A_s_ and A_f_ of TMT, the number of valence electrons per atom e_v_/a, and their concentration c_v_, average grain size <d> and mechanical properties of Cu–Al–Ni-based alloys.

Alloy, at.%	A_s_, K	A_f_, K	e_v_/a	c_v_	<d>, μm	σ*_m_*, MPa	σ_u_, MPa	δ, %	Ψ, %
Cu_78_Al_19_Ni_3_	903	943	9.45	0.364	60	280	780	15	0.5
Cu_77_Al_20_Ni_3_	893	933	9.33	0.363	80	260	520	10	0.5
Cu_75_Al_22_Ni_3_	793	833	9.21	0.362	130	260	490	6	0.5
Cu_73_Al_24_Ni_3_	673	723	9.05	0.360	350	260	450	5	0.5
Cu_71_Al_26_Ni_3_	513	560	8.89	0.358	750	200	390	4	0.5
Cu_69_Al_28_Ni_3_	283	333	8.77	0.357	1000	120	250	3	0.5

The chemical composition of the basic elements varied within 0.1 at.% and O and C were within 0.07–0.1 wt.%.

**Table 5 materials-12-02616-t005:** Chemical composition, temperatures of TMTs B2 ↔ B19′, the number of valence electrons per atom e_v_/a and their concentration c_v_.

Alloy, at.%	B19′				e_v_/a	c_v_
	M_s_, K	M_f_, K	A_s_, K	A_f_, K		
Ti_50_Ni_49.5_Cr_0.5_ [38] *	292	264	302	329	6.98	0.279
Ti_49_Ni_50_Cr_1_ [38] *	237	215	252	268	7.02	0.281
Ti_49_Ni_50_V_1_ [38] *	318	284	327	348	7.01	0.280
Ti_48_Ni_50_V_2_ [38] *	309	285	316	340	7.02	0.281
Ti_47_Ni_50_V_3_ [38] *	293	271	300	318	7.03	0.281
Ti_45_Ni_50_V_5_ [38] *	283	269	286	297	7.05	0.281
Ti_44_Ni_50_V_6_ [38] *	279	261	282	295	7.06	0.282
Ti_50_Ni_45_Cu_5_ [38] *	345	317	340	368	7.05	0.281
Ti_50_Ni_40_Cu_10_ [38] *	306	285	300	316	7.10	0.283
Ti_50_Ni_45_Cu_5_ [43] **	309	295	311	330	7.05	0.281
Ti_50_Ni_40_Cu_10_ [43] **	292	179	192	200	7.10	0.283

The content of C and O was less than 0.01 wt.% (*) and the content of O and C was nearly 0.1 wt.% (**).

**Table 6 materials-12-02616-t006:** The magnitudes of the volume effect (V/V) and lattice parameters of the B2 austenite and of the B19, and B19′ martensites near the M_s_ temperature of the binary and ternary TiNi-based alloys.

Alloy, at.%	ΔV/V, %	B2	Martensite B19′ (B19)			
		a, nm	a, nm	b, nm	c, nm	β, °
Ti_50_Ni_50_	−0.13	0.3015	0.2890	0.4120	0.4630	96.8
Ti_49.4_Ni_50.6_	−0.31	0.3013	0.2876	0.4132	0.4622	97.0
Ti_50_Ni_49_Co_1_	−0.24	0.3014	0.2882	0.4115	0.4644	97.3
Ti_50_Ni_46_Co_4_	−0.86	0.3013	0.2874	0.4108	0.4630	97.2
Ti_50_Ni_45_Co_5_	0.49	0.3012	0.2871	0.4104	0.4697	97.1
Ti_50_Ni_45_Cu_5_ [46]	−0.5	0.3027	0.2903	0.4147	0.4613	96.2
Ti_50_Ni_45_Cu_7.5_ [46]	0.36	0.3030	0.2911	0.4228	0.4517	90
Ti_50_Ni_45_Cu_7.5_ [46]	−0.27	0.3030	0.2907	0.4170	0.4596	95.5
Ti_50_Ni_40_Cu_10_ [46]	−0.36	0.3031	0.2901	0.4249	0.4515	90
Ti_50_Ni_40_Cu_10_ [46]	−1.44	0.3031	0.2872	0.4192	0.4577	95.2
Ti_50_Ni_37.5_Cu_12.5_ [46]	−0.14	0.3034	0.2896	0.4252	0.4514	90
Ti_50_Ni_35_Cu_15_ [46]	−0.44	0.3043	0.2899	0.4260	0.4516	90
Ti_50_Ni_30_Cu_20_ [46]	−0.53	0.3046	0.2900	0.4264	0.4512	90
Ti_50_Ni_41_Pd_9_ [47]	−0.78	0.3047	0.2846	0.4304	0.4583	90
Ti_50_Ni_40_Pd_11_ [47]	−0.93	0.3050	0.2830	0.4314	0.4604	90
Ti_50_Ni_32_Pd_18_ [47]	−0.68	0.3056	0.2820	0.4343	0.4628	90
Ti_50_Ni_30_Pd_20_ [47]	−0.58	0.3051	0.2820	0.4340	0.4613	90
Ti_50_Ni_25_Pd_25_ [47]	−0.53	0.3063	0.2807	0.4361	0.4667	90
Ti_50_Au_50_ [48]	−0.11	0.3220	0.294	0.463	0.490	90

**Table 7 materials-12-02616-t007:** The martensite lattice parameters of ternary TiNi- and Ni_2_MnGa-based alloys.

Alloy, at.%	a, nm	b, nm	c, nm	β, °
Ti_50_Ni_5_Pd_45_	0.2803	0.454	0.4794	90
Ti_50_Ni_48_Pt_2_	0.2892	0.4135	0.4643	96.6
Ti_50_Ni_42_Pt_8_	0.2821	0.4292	0.4585	90
Ti_50_Ni_42_Pt_8_	0.2873	0.4211	0.4627	95.6
Ti_50_Ni_25_Pt_25_	0.2765	0.4483	0.4744	90
Ti_50_Ni_47_Au_3_	0.2907	0.4136	0.4653	96.6
Ti_50_Ni_45_Au_5_	0.2907	0.4159	0.4666	96.5
Ti_50_Ni_38_Au_12_	0.2876	0.4319	0.46203	90
Ti_50_Ni_25_Au_25_	0.2864	0.4493	0.47492	90
Ni_50_Ti_47_Zr_3_	0.292	0.4114	0.467	97.8
Ni_50_Ti_45_Zr_5_	0.293	0.411	0.472	98.5
Ni_50_Ti_40_Zr_10_	0.298	0.410	0.478	100.3
Ni_50_Ti_35_Zr_15_	0.303	0.409	0.487	101.8
Ni_50_Ti_32_Zr_18_	0.305	0.408	0.492	102.5
Ni_50_Ti_30_Zr_20_	0.307	0.408	0.495	103.7
Ni_50_Ti_38_Hf_12_	0.3006	0.4116	0.4803	101.2
Ni_50_Ti_35_Hf_15_	0.3025	0.4096	0.4826	102
Ni_50_Ti_32_Hf_18_	0.3051	0.4090	0.4850	102.5
Ni_50_Ti_30_Hf_20_	0.3063	0.4083	0.4890	103
Ni_50_Mn_25_Ga_25_	0.555	0.555	0.670	90
Ni_53_Mn_22_Ga_25_	0.4142	0.556	2.953	90
Ni_54_Mn_21_Ga_25_	0.4232	0.550	2.937	93.3
Ni_54.75_Mn_20.25_Ga_25_	0.549	0.549	0.6478	90
Ni_55.25_Mn_19.75_Ga_25_	0.548	0.548	0.653	90
Ni_56.75_Mn_18.25_Ga_25_	0.546	0.546	0.653	90
Ni_57.5_Mn_17.5_Ga_25_	0.542	0.542	0.660	90
Ni_58.25_Mn_16.75_Ga_25_	0.543	0.543	0.663	90
Ni_59_Mn_16_Ga_25_	0.543	0.543	0.663	90
Ni_59.75_Mn_15.25_Ga_25_	0.543	0.543	0.663	90

**Table 8 materials-12-02616-t008:** Mechanical properties of fine-grained (FG) alloys at room temperature.

Alloy, at.%	σm, MPa	σy, MPa	σu, MPa	σr, MPa	δ, %	er, %
Ti_50_Ni_50_	200	850	1200	650	40	6
Ti_50_Ni_48_Cu_2_	150	670	850	520	20	5
Ti_50_Ni_45_Cu_5_	120	580	750	460	18	4
Ti_50_Ni_40_Cu_10_	100	530	750	430	25	3
Ti_50_Ni_35_Cu_15_	70	490	720	420	20	3
Ti_50_Ni_30_Cu_20_	60	500	720	440	20	3
Ti_50_Ni_25_Cu_25_	60	500	740	440	20	3
Ti_50_Ni_49_Fe_1_	200	720	1000	520	24	5
Ti_50_Ni_47_Fe_3_		650	950		18	
Ti_50_Ni_49_Co_1_	120	740	1100	620	18	5
Ti_50_Ni_48_Co_2_	160	740	1100	580	18	4
Ti_50_Ni_47_Co_3_	240	710	1100	470	16	3
Ti_50_Ni_45_Co_5_		650	1090		15	
Ti_50_Ni_43_Co_7_		630	1080		14	
Ti_50_Ni_10_Pd_40_	380	850	1010	470	11	5

The content of O and C in the range of 0.07–0.1 wt.%.

**Table 9 materials-12-02616-t009:** Mechanical properties of high-purity alloys Ti_49.4_Ni_50.6_, quenched and subjected to ECAP (1), HPT (2) and annealing.

Alloy, at.%	<d>, m	σm, MPa	σy, MPa	σu, MPa	σr, MPa	δ, %	εr, %
Ti_49.4_Ni_50.6_	60	130	630	1600	500	75	5
Ti_49.4_Ni_50.6_ ^(1)^	0.40	250	1200	1600	950	60	8
Ti_49.4_Ni_50.6_ ^(2)^	0.03	460	1800	2100	1340	16	10
Ti_49.4_Ni_50.6_ ^(2)^	0.05	370	1500	1670	1130	23	13
Ti_49.4_Ni_50.6_ ^(2)^	0.10	200	1120	1250	920	30	16
Ti_49.4_Ni_50.6_ ^(2)^	0.60	250	900	1210	650	75	16

*—impurity content, C—0.0372 wt.%, O—0.0167 wt.%, N—0.003 wt.%, S—0.0001 wt.%.

**Table 10 materials-12-02616-t010:** Mechanical properties of RQM alloys of the Ti–Ni–Cu system.

Alloy, at.%	<d>, m	σm, MPa	σy, MPa	σu, MPa	σr, MPa	δ, %	εr, %
Ti_50_Ni_25_Cu_25_	1.0	60	680	850	620	12	5
Ti_50_Ni_25_Cu_25_	0.8	70	720	900	650	12	4
Ti_50_Ni_25_Cu_25_	0.5	80	1070	1200	990	11	4
Ti_50.5_Ni_25_Cu_24.5_	0.3	80	1050	1170	970	9	3
Ti_50.5_Ni_24.5_Cu_25_	0.3	90	1150	1380	1060	10	3
Ti_49.5_Ni_25_Cu_25.5_	0.3	80	950	1120	870	10	3
Ti_49_Ni_25_Cu_26_	0.2	90	1150	1380	1060	10	3
Ti_51_Ni_24_Cu_25_	0.2	100	1160	1300	1060	9	3
Ti_51_Ni_25_Cu_24_	0.2	90	1200	1550	1110	10	3

The chemical composition of the basic elements varied within 0.1 at.% and O and C were within 0.07–0.1 wt.%.

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
