# Peer review of "Design and Development of Ti–Ni, Ni–Mn–Ga and Cu–Al–Ni-based Alloys with High and Low Temperature Shape Memory Effects"

_materials, 2019, doi:10.3390/ma12162616_

Round 1

Reviewer 1 Report

The manuscript focuses on designing and developing multicomponent alloys with thermoelastic martensitic transformations in order to improve their strength and ductile properties. The author provided plenty of valuable experimental data and obtained some interesting results. The manuscript can be considered for publication in the journal after experiencing the major revisions.

1. The title is not appropriate. Ductile and plastic possess the almost same meaning and thus they are not used simultaneously. In addition, the title does not cover copper alloys.

2. In the manuscript, the authors deal with severe plastic deformation. However, the authors do not give any current overview with respect to severe plastic deformation of NiTi-based shape memory alloys in the introduction or text section. Some literatures should be cited, for high pressure torsion (HTP), equal channel angular extrusion (ECAE), local canning compression, plane strain compression, cold rolling and cold drawing. For example (but are not limited to )

(1)Zhang et al. Deformation mechanism of NiTi shape memory alloy subjected to severe plastic deformation at low temperature. Materials Science and Engineering A, 559, 2013, 607−614.

(2)Shahmir et al. Shape memory effect in nanocrystalline NiTi alloy processed by high-pressure torsion. Mater. Sci. Eng., A 2015, 626, 203-206.

3. On the whole, the references are very old. Some new references should be provided. For instance, in the manuscript, the author mentioned the twins, so some important references should be cited for the purpose of comparison. For example, (but are not limited to)

(1) Hu et al. Transformation twinning and deformation twinning of NiTi shape memory alloy. Materials Science and Engineering A, 2016, 660: 1–10.

4. English language must be improved. Some sentences are difficult to be understood. For example (but are not limited to)

(1) Page 19, Line 425, “This indicates a predominantly intergranular fracture mechanism along mainly by large-angle boundaries of LG alloys.” The sentence seems to be incomplete and it is difficult to be understood. “along mainly”?

(2) Page 19, Line 425, “The elements Zr and Hf from common with Ti subgroup IVA of the Periodic Table of elements have the increased solubility (up to 25-30 at. %) by means of substitution instead of Ti in the quasi-binary alloys NiTi-NiMe.” is also a bad sentence. “from common”?

(3) There are still too many grammar errors.

Author Response

Thank You for useful advices on our article «Design and Development of High-Strength and Ductile Intermetallic Alloys of Ti-Ni, Ni-Mn-Ga and Cu-Al-Ni- based Systems with High and Low Temperature Shape Memory Effects». Manuscript ID: materials-560417.

We have tried to take into account all your critical remarks in the revised text (attached below):

– the title has been changed to take into account the wishes of experts;

– the text has been supplemented by the citation of the papers for last ten years, including the recommendations of experts;

– the problems of severe plastic deformation have been discussed in more detail, with adding references;

– we have tried to identify the common behavior of other alloys with the titanium nickelide-based alloys, including the correlation between the temperatures of the start of martensitic transformations and the electron characteristics;

– the differences in the alloys typical of the high-temperature and low-temperature thermo-elastic transformations have been distinguished;

– captions of the tables and figures have been updated.

Sincerely Yours, authors!

Reviewer 2 Report

The work presents an extensive collection of data referring to the properties and characteristics of a vast set of alloys, belonging to three different families, with the common feature of presenting thermoelastic martensitic transformation and the associated shape-memory properties.  The information it contains already makes the work valuable. However, there are several aspects of the work that I think deserve to be reconsidered:

- In the first place, although many data are included in the tables for Ni-Mn-Ga based and Cu-Al-Ni-based alloys, analysis in the text hardly refers to these alloys, so that appropriate comparison cannot be established (if this is the intention of the authors). I suggest revising the discussion to ensure that the role of such alloys is clear.

- Secondly, the discussion about dependence on the concentration transformation temperatures is excessively qualitative. I wonder if you could not establish the master lines of this unit through graphics or, at least, quote them explicitly.

- Thirdly, although the title explicitly mentions the concept of high temperature, there is hardly any reference to it in the text. In fact, the tables include among the studied alloys varied transformation temperatures, and I cannot find a discussion about what is the domain of temperatures of interest and what are the implications of transforming at high temperatures.

Otherwise, I also suggest that the authors ensure to correctly define in the text or on the tables captions the magnitudes that appear in them.

Author Response

(The authors gave the same response as above.)

Round 2

Reviewer 1 Report

The authors have given good response to my comments. The manuscript can be recommended for publication in the journal.

Author Response

Thank You for useful advices on our article «Design and Development of High-Strength and Ductile Intermetallic Alloys of Ti-Ni, Ni-Mn-Ga and Cu-Al-Ni- based Systems with High and Low Temperature Shape Memory Effects». Manuscript ID: materials-560417.

Sincerely Yours, authors!